# Molecular Recognition Patterns between Vitamin B12 and Proteins Explored through STD-NMR and In Silico Studies

**DOI:** 10.3390/foods12030575

**Published:** 2023-01-28

**Authors:** Ruchira Ghosh, Donald S. Thomas, Jayashree Arcot

**Affiliations:** 1Food and Health, School of Chemical Engineering, UNSW Sydney, Sydney, NSW 2052, Australia; 2NMR Facility, UNSW Sydney, Sydney, NSW 2052, Australia

**Keywords:** cyanocobalamin, aqua cobalamin, STD–NMR, interaction, molecular docking, proteins

## Abstract

Ligand–receptor molecular recognition is the basis of biological processes. The Saturation Transfer Difference–NMR (STD–NMR) technique has been recently used to gain qualitative and quantitative information about physiological interactions at an atomic resolution. The molecular recognition patterns between the cyanocobalamin (CNBL)/aqua cobalamin (OHBL) and different plant and animal proteins were investigated via STD–NMR supplemented by molecular docking. This study demonstrates that myoglobin has the highest binding affinity and that gluten has the lowest affinity. Casein also shows a higher binding affinity for cyanocobalamin when compared with that of plant-based proteins. STD–NMR results showed the moderate binding capability of casein with both CNBL and OHBL. Computer simulation confirmed the recognition mode in theory and was compared with the experiments. This work is beneficial for understanding the binding affinity and biological action of cyanocobalamin and will attract researchers to use NMR technology to link the chemical and physiological properties of nutrients.

## 1. Introduction

Vitamin B12 (cobalamin) also known as the anti-pernicious anemia factor, is a structurally complex water-soluble B vitamin. Biologically, vitamin B12 works as a coenzyme and participates in different metabolic processes [1]. However, the human body cannot synthesize vitamin B12, and it is only found in animal food sources, such as meat, fish, milk, etc. Thus, it is generally used as a fortificant in different plant-based foods and dietary supplements to fulfill the requirements. Vitamin B12 deficiency is associated with neurological metabolic and haematological disorders [1,2,3]. Animal-based foods typically provide the necessary daily amount of 2.4 g, equivalent to 1.8 nmol [3,4]. Cow milk, for instance, provides 0.6 to 3.6 nmol of Cbl/L [5,6], resulting in high bioavailability of the vitamin [5]. Caseins in bovine milk bind high quantities of HOCBL, due to the formation of casein–Cbl co-ordination bonds [7]. The casein–Cbl bond is sufficiently stable to pass through the stomach’s acidic medium, indicating that casein and its peptides play a role in Cbl absorption by intestinal cells [7]. This indicates that the vitamin B12 binding proteins are well digested, which would otherwise limit absorption [8].

Nuclear magnetic resonance (NMR) provides unrivaled benefits in researching molecular recognition mechanisms, as compared to traditional approaches, such as isothermal calorimetric titration. NMR is gaining popularity in biology as it enables the real-time investigation of biological systems within its working conditions [9]. It is a specialized tool for studying molecular interactions in a solution, and it has become an important analytical method used to analyze molecular recognition and obtain information about the interactions of small ligands with essential biological macromolecules (proteins or nucleic acids) [10]. The saturation transfer difference NMR (STD–NMR) technique has been used to describe ligand–receptor complexes for several years. The STD–NMR experiment depends on the fact that there is an interchange between the bound and unbound ligand states for a weak-binding ligand. In an STD experiment, a spectrum in which the protein is selectively saturated (on-resonance spectrum-I_SAT_) with signal intensities is subtracted from the one recorded without protein saturation (off-resonance spectrum), with signal intensities of I_0_. In the difference spectrum (I_STD_ = I_0_ − I_SAT_), only the ligand(s) signals that received saturation transfer from the protein will remain. Low sensitivity to very small amounts of ligand, a prolonged experimental time, and incompatibility with high-affinity compounds are the main limitations of STD–NMR. However, STD–NMR has a lower susceptibility to false positives [11] compared to other techniques, including receptor-based NMR techniques, and there is no upper size threshold for the receptor. STD–NMR has the capacity to not only offer information on binding affinity between ligands and their target protein(s), but also to provide vital insights into the protein architectural changes that may occur when they bind to their ligand(s) [12].

In the present work, we set out to study the interaction between cyanocobalamin/aqua cobalamin and several proteins using STD–NMR. Although many studies indicate casein–OHBL binding, it is not yet clear how cyanocobalamin/aqua cobalamin bind with different proteins or if plant-based proteins can bind with cyanocobalamin/aqua cobalamin. This study was undertaken to enhance the understanding of the nature of different animal and plant-based proteins in terms of Cbl binding, because of its relevance to the bioavailability and quantity of the vitamin in plant-based products. All animal protein sources are naturally rich in vitamin B12, and the chosen plant proteins are from staple foods consumed by a wide range of the population. The hypothesis of this study was that an equal concentration of cyano and aqua cobalamin shows different interactions with different animal and plant proteins. Moreover, any plant protein with a similar interaction pattern, compared with that of an animal protein, can be a potential fortification medium. The results describe the interactions between cyanocobalamin/aqua cobalamin and different plant-based protein fractions compared to different animal proteins.

## 2. Hypotheses

An equal concentration of cyano and aqua cobalamin will form different interactions with different animal and plant proteins.

## 3. Materials and Methods

### 3.1. Reagents

Cyanocobalamin, aqua cobalamin, casein, egg albumin, myoglobin, and gluten were purchased from Sigma-Aldrich (Sydney, Australia). D_2_O was purchased from UNSW chemical stores. Brown rice protein (BRP) isolate (90% purity) and yellow pea protein (YPR) (90% purity) isolate were acquired from Axiom Foods, Inc. (through Pacific Resources Intl., Sydney, Australia). All reagents were of analytical grade and were used without further purification. The stock solutions of Casein, egg albumin, myoglobin, gluten, rice protein, pea protein (80 μmol/L), and cyanocobalamin and aqua cobalamin (0.07, 0.22, 0.37, 3.69, 7.38, 8.86, 22.14, 36.90, 51.66, 66.42, 88.56 μmol/L) were prepared using D_2_O.

### 3.2. Characterization of Cobalamin

To assess the compound stability and solubility, ^1^H-NMR spectra were acquired on a 600 MHz Bruker Avance III HD spectrometer using a 1 µmol/L solution of the cyanocobalamin in 100% deuterated water with a pH of 7.4 and a temperature of 298 K. Full compound characterization at the optimized conditions (pH 7.4, 100% D_2_O, and 298 K) was performed for both cyano and aqua cobalamin and required further acquisition and assignment of the Nuclear Overhauser Effect Spectroscopy (NOESY) spectra.

### 3.3. Saturation Transfer Difference via Nuclear Magnetic Resonance (STD–NMR)

With a dissociation constant (K_D_) ranging from M to mm, interactions with binding events that take place during the fast exchange of the saturation transfer are identified by the term “saturation transfer difference” (STD) [13]. When analyzing compounds that affect protein activity rather than totally blocking it [14] or when screening potential ligands for fragment-based drug design, ligands in this range are particularly relevant [15,16].

NMR data were gathered using a Bruker AVANCE III HD (Bruker, Germany) that ran based on 600 MHz fitted with a 5 mm TCI helium cryoprobe and refrigerated sample jet autosampler. The STD studies were carried out using Bruker pulse sequence STDDIFFESGP.3 with protein suppression via a spin lock filter [13,17]. All proteins were exposed to radiation at a chemical shift of 0 ppm during on-resonance irradiation and 40 ppm during off-resonance irradiation. The following settings were used to obtain the spectra: a relaxation delay (D1) of 4 s, saturation time (D20) of 3 s, number of blocks of TD size (NBL) of 2, a spectrum window of 16 ppm, time domain size of 32,768, and a scan number of 128. Samples were produced in 5 mm NMR tubes (Norell^®^ Sample Vault Series) using 100% D_2_O. Calculating the STD impact resulted in:STD = (Ioff − Ion)/Ioff
where Ioff and Ion are the integrals of each ligand signal in the off- and on-resonance spectra, respectively. Data were collected in the STD experiment with each protein ligand individually. Bruker’s Topspin 4.0.7 software was used for all NMR data processing.

### 3.4. Molecular Docking

Molecular docking simulation at 298 K was obtained using CB-Dock2 software freely available at http://cao.labshare.cn/cb-dock/ (accessed between 1 May to 31 August 2022). CB-Dock2 is an upgraded version of the protein–ligand blind docking program that incorporates the CB-Dock server’s curvature-based cavity identification process, as well as the Auto Dock Vina-based molecular docking procedure. This online platform was used between May and August 2022. The single crystal model of different proteins [casein (PDB ID.: 5VYA), egg albumin (Uniport ID.: P19121), bovine myoglobin (Uniport ID.: P02192), gliadin from wheat gluten (PDB ID.: 4GG6), rice protein glutelin (Uniport ID.: Q09151), pea protein (PDB ID.: 3KSC)] as a receptor was from the Protein Data Bank, and the 3D structure of cyano and aqua cobalamin was sourced from PubChem. Different proteins and cyano and aqua cobalamin were optimized to ensure the correct proton state and system energy minimization. The top docking results were used for further analysis. Final visualization and observations were performed using discovery studio visualiser.

## 4. Results and Discussion

### 4.1. Cyanocobalamin Structure after the ^1^H Scan in NMR

There are distinctive structural properties of vitamin B12, including tetrapyrrole rings, a 5,6-dimethylbenzimidazole group, and the reactive center cyano group around the core atom of cobalt [18,19,20].

Cyanocobalamin was split into four pyrrole rings (A–D), seven districts (a–g), and one big outer ring for ease of description. The signal peaks of cyanocobalamin in the ^1^H NMR spectra are allocated in Figure 1 based on the structure of the cyanocobalamin and previous reports, which is crucial to assure the correctness to assess the results that follow. Like that of cyanocobalamin, the aqua cobalamin structure was also analyzed using NMR. Aqua cobalamin was split into four pyrrole rings (A–D), seven districts (a–g), and one big outer ring for ease of description. The signal peaks of aqua cobalamin in the ^1^H NMR spectra are allocated in Figure 2 based on the structure of aqua cobalamin and previous reports, which is crucial to assure the correctness to assess the results that follow. The presence of the -OH group instead of cyanide group, which caused a slight shift in the different peak of the NMR spectra.

### 4.2. Interaction between Vitamin B12 and Protein Identified Using STD–NMR

STD–NMR experiments revealed the water-soluble parts of every protein interaction with cobalamin. Off-resonance and difference spectra were observed for each protein independently. Cyanocobalamin had a distinct signal observed in the STD difference spectra at 0.3, 1.84, 2.18, 2.48, 2.49, 6.00, 7.00, and 7.19 ppm with all proteins. Calculating the relative ratio between off-resonance and the difference spectrum signal area is possible to assess the binding epitopes and thus the degree of proximity between ligand atoms and the protein-binding site. The closer the ligand atom is to the protein, the more saturation it will receive and as a result, the higher its signal intensity will be in the difference spectrum. A similar distinct signal was also observed in the STD difference spectra for other proteins. Because of the NMR spectrum shift, aqua cobalamin has a distinct signal observed in the STD difference spectra at 0.437, 1.85, 2.4, 2.55, 2.64, 6.15, 6.45, and 7.08 ppm with all different proteins.

### 4.3. Binding Capability Judgment Based on Cyanocobalamin

Binding capability is an important parameter when studying ligand–protein interactions because it affects the biological function of a protein and therefore the bioavailability of a ligand. In this work, STD–NMR experiments were used to calculate the binding constant quantitatively using Equations (S3) and (S4). Given the severe signal overlap, the single peak was selected and the scale values between the off-resonance and STD spectra were used for precise calculations. Nine groups of peaks, including C20, C25, C35, C53, C10, B (10/11), B2, B4, and B7 were selected to classify the groups of cobalamins relevant for interactions with both plant and animal proteins. The binding constant K_α_ values evolved from K_D_ for these nine signals(Table 1).

The average values of K_α_ were calculated to represent the binding capability of the cyanocobalamin and gluten, myoglobin, casein, egg albumin, pea, and rice proteins, which were 4.26 × 10^3^, 1.09 × 10^4^, 9.76 × 10^3^, 2.03 × 10^4^, 2.94 × 10^4^, and 3.44 × 10^4^ L/mol at room temperature. Similarly, the K_α_ values of aqua cobalamin and gluten, myoglobin, casein, egg albumin, pea, and rice proteins were 3.9 × 10^3^, 4.3 × 10^4^, 3.5 × 10^4^, 3.9 × 10^4^, 3.9 × 10^4^ and 1.1 × 10^4^, respectively (Table 2). Usually, the values of K_α_ between small molecules and a protein reflects the strength of binding, where less than 1 × 10^4^ L/mol is of low affinity, 10^4^~10^5^ L/mol is medium affinity, and above 10^6^ L/mol is high affinity [21]. Moderate binding ability indicates that it is a suitable system with a binding–dissociation rate for STD–NMR studies [11,22]. Comparing different proteins, myoglobin showed the highest, whereas gluten showed the lowest, binding affinity for the cyanocobalamin. Egg albumin showed the highest, and gluten showed the lowest, binding affinity for aqua cobalamin.

The requirement for Cbl from foreign sources is universal across the animal kingdom, while it is different in herbivores and carnivores. Microorganisms that inhabit the digestive system of ruminants manufacture the vitamin through the bacterial fermentation of cellulose [23]. Binding interactions of vitamin B12 with different proteins will help to understand the potency of those proteins as a transporter for vitamin B12 [8,24]. In the current study, we observed that animal protein binding affinity is significantly higher than that of plant proteins. This could be the reason for different bio-availabilities of vitamin B12 from various sources. The tetrapyrrole rings of Aqua cobalamin and 5,6-dimethyl benzimidazole group of cyanocobalamin shows higher binding affinity compared to the rest of the chemical structure.

### 4.4. Binding Conformation Simulation

Molecular docking is a powerful tool to validate the 3D model complexes by comparing experimental data with those predicted from the model [25,26]. The binding conformation of both aqua cobalamin and cyanocobalamin and different proteins predicted using CB- Dock 2 and discovery studio visualizer is presented in Appendix A (Appendix A).

Both the hydroxy and the cyanocobalamin molecules were inserted into the interior of the protein site maintained by the intermolecular non-covalent interaction forces. Hydroxy and cyanocobalamin stretched along chain A of casein, egg albumin, and myoglobin, chain A of rice glutelin, the D, E, F chain of Pro legumin from pea protein, and the G, H chain of gliadin from wheat. The easily dissociated groups of hydroxy and cyanocobalamin played a key role in the formation of hydrogen bonds, and the substituent on the tetra pyridine ring of the cyanocobalamin was important for the binding with proteins [20]. The binding conformation predicted via computer simulation confirmed the experimental results.

The main binding forces of cyanocobalamin to casein were hydrogen bond, π-sigma, π-alkyl, and π-π interactions. The residues involved in casein (−5.9 kcal/mol) and cyanocobalamin binding were found to be ASP184, VAL186, ILE187, ARG189, PRO214, THR219, ALA220, GLU223, ASP284, ILE351, LYS358, TYR359, PRO389, ASP390, LEU393, and ASP394 (Table 3). The protein–ligand interaction is shown in a figure (Appendix A) and demonstrates that the CNBL interaction is stabilised by twelve π-π and π-alkyl interactions and fourteen hydrogen bonds. The oxygen atom on P=O of cyanocobalamin formed two hydrogen bonds with the OH group of TYR 359 of casein with a distance of 2.63 Å and 3.18 Å. NMR results demonstrated that B10 and B11 of cyanocobalamin (DMB) showed the highest binding affinity with casein. The NH group in the Corrin district of cyanocobalamin was bound with the oxygen atom and OH group of Asp 390 with a hydrogen bond distance of 2.53 and 2.54 Å. Other NH groups of cyanocobalamin were bound with the OH group of LYS358 and THR219 with a hydrogen bond distance of 2.74 and 2.86 Å. On the other hand, the main binding forces of aqua cobalamin to casein were only hydrogen bonds. The residues involved in casein (−4.4 kcal/mol) and aqua cobalamin binding were determined to be SER344, VAL345, ARG346, VAL349, ARG353, VAL373, GLN377, GLY629, ASN633, ASP634, GLU635, GLU792, GLU793, ARG794, PHE795, GLU796, GLN797, and ASN798. The protein–ligand interaction is shown in a figure (Appendix A) and demonstrates that the OHBL interaction is stabilised by eight hydrogen bonds. NMR results demonstrated that C25 and C35 of aqua cobalamin pyrrole rings showed the highest binding affinity with casein. The CONH_2_ group in the aqua cobalamin was bound with the COOH group of ASN798, GLU796, GLU792, and PHE795 with hydrogen bond distances of 2.80, 2.52, 2.84, and 2.80 Å. The readily dissolved groups in cyanocobalamin were crucial for the creation of π -bonds, and the substituent on the 5,6-dimethylbenzimidazole was crucial for the binding to casein.

The residues involved in myoglobin (−5.1 kcal/mol) and cyanocobalamin binding were found to be PHE44, LYS46, PHE47, ASP61, HIS65, THR68, ALA72, ALA85, HIS89, and HIS98. Vitamin B12 and myoglobin interaction is stabilised by five hydrogen bonds and nine π-π and, π-alkyl interactions. The oxygen atom on P=O of cyanocobalamin formed hydrogen bonds with the OH group of THR68 of casein at a distance of 1.96 Å. NMR results demonstrated that B10, B11, B7, and B4 of cyanocobalamin (DMB) showed the highest binding affinity for myoglobin. The imidazole group of His98, benzyl group of PHE47 and PHE44, and CH group of Lys46 were crucial for the creation of π-π and π-alkyl bonds, with 5,6-dimethylbenzimidazole being crucial for binding with myoglobin. The residues involved in myoglobin (−5.2 kcal/mol) and aqua cobalamin binding were determined to be ARG32, LEU33, GLY36, HIS37, LYS103, GLU106, PHE107, ILE108, ASP110, ALA111, ILE113, HIS114, HIS117, and GLN129(Table 3). The vitamin B12 and myoglobin interaction is stabilised by 10 hydrogen bonds and 4 π-π interactions. The CONH_2_ group of pyrrole rings formed hydrogen bonds with the OH group of ARG32, PHE107, ASP110, and GLN129 of myoglobin with distances of 3.07, 2.68, 1.90, and 2.10 Å. NMR results demonstrated that pyrrole rings of aqua cobalamin showed the highest binding affinity with myoglobin. The imidazole group of His114 and NH_2_ group of ILE113 were crucial for the creation of π- π bonds, with the pyrrole ring being crucial for the binding with myoglobin.

The residues involved in chicken egg albumin (−6.5 kcal/mol) and cyanocobalamin binding were determined to be VAL222, LYS223, TYR246, MET319, GLU320, GLU322, PHE323, LYS457, LYS464, LEU468, ARG472, ALA475, TYR480, ILE483, and VAL484. The vitamin B12 and chicken egg albumin interaction is stabilised by 11 hydrogen bonds and 13 π-π and π-alkyl interactions. The oxygen atom on P=O of cyanocobalamin formed hydrogen bonds with the NH_2_(NZ) group of LYS357 and OH group (adjacent phenyl group) of TYR480 in casein at distances of 3.14 and 2.77 Å. 5,6-Dimethylbenzimidazole is highly interactive with the β-CH group VAL 222 (4.04 Å), γ CH2 group LYS457 (4.53 Å), γ CH2 group ILE483 (5.23 Å), β-CH group VAL484 (4.12 Å) and phenyl ring TYR 480 (5.05 Å) creating π-π and π-alkyl bonds. On the other hand, residues involved in chicken egg albumin (−5.6 kcal/mol) and aqua cobalamin binding were determined to be GLU43, GLU44, LYS47, MET51, TYR63, SER67, LYS68, VAL70, LYS71, ASP75, GLN78, ASN158, VAL160, SER161, and HIS165. The aqua cobalamin and chicken egg albumin interaction is stabilised by 10 hydrogen bonds and 5 π-π and π-alkyl interactions.

The residues involved in gliadin from wheat gluten (−7.1 kcal/mol) and cyanocobalamin binding were determined to be PRO46, ASP174, VAL177, LEU46, ARG123, VAL168, LEU170, PRO187, GLN188, PRO189, LEU190, and LYS191. The vitamin B12 and wheat protein gliadin interaction is stabilised by nine hydrogen bonds and four π-alkyl interactions. The oxygen atom in P=O of cyanocobalamin formed hydrogen bonds with the NH_2_(η-position) group of ARG123 in glutelin at a distance of 3.24 Å. The 5,6-dimethylbenzimidazole group of cyanocobalamin forms π-bonds with VAL177 at a distance of 4.91 Å. On the other hand, the corrin structure of cyanocobalamin is stabilised with LEU46, ASP174, GLN188, and PRO189 with hydrogen bonds and PRO 46, LYS191, and LEU190 with π-alkyl interactions in the glutelin protein. The residues involved in gliadin from wheat gluten (−7.5 kcal/mol) and aqua cobalamin binding were found to be HIS10, GLN44, LEU101, PHE103, PRO120, ARG123, ASP166, HIS167, VAL168, GLU169, LEU170, SER171, GLU178, VAL179, HIS180, SER181, GLY182, VAL183, CYS184, THR185, ASP186, PRO187, GLN188, PRO189, LEU203, and TYR228. The aqua cobalamin and wheat protein gliadin interaction is stabilised by 14 hydrogen bonds and 12 π-alkyl interactions. The oxygen atom in P=O of aqua cobalamin formed hydrogen bonds with the NH_2_(η-position) group of ARG123 in glutelin at a distance of 1.87 Å. Pyrrole rings of aqua cobalamin form H-bonds with VAL183, the imidazole ring of HIS180, and HIS10, LEU170, PRO187, and VAL168 at distances of 2.76, 2.85, 3.01, 3.07, 2.80, and 3.07 Å.

The residues involved in glutelin protein from rice (−8.1 kcal/mol) and cyanocobalamin binding were found to be Arg199, His200, Arg201, Phe203, Phe204, Ile211, Leu215, Glu219, Asp222, Ser224, Asn226, and Val227. The vitamin B12 and rice-glutelin interaction is stabilised by 14 hydrogen bonds and 7 π-alkyl interactions. The oxygen atom on P=O of cyanocobalamin formed hydrogen bonds with the NH_2_(η-position) group of ARG201 in glutelin at a distance of 5.97 Å. The 5,6-dimethylbenzimidazole group of cyanocobalamin forms π-bonds with HIS200, ILE211, LEU215, and PHE204. On the other hand, the corrin structure of cyanocobalamin is stabilised with VAL227, ASN226, and ASP222 with hydrogen bonds and PHE203 with π-alkyl interactions in the glutelin protein. The residues involved in glutelin protein from rice (−6.7 kcal/mol) and aqua cobalamin binding were determined to be TRP34, GLN35, SER36, SER37, ARG38, ARG39, GLY40, SER41, GLU44, CYS45, ARG46, PHE47, CYS78, THR79, TYR190, ILE371, ASN372, HIS374, ASN391, GLN411, HIS412, and HIS413. The aqua cobalamin and rice-glutelin interaction is stabilised by 16 hydrogen bonds and 6 π-alkyl interactions. The 5,6-dimethylbenzimidazole group of aqua cobalamin formed π-π interactions with the imidazole group of HIS412 in glutelin at a distance of 5.37 Å. The pyrrole ring aqua cobalamin forms H-bonds with CYS78 (2.36 Å), GLN411 (2.84 Å), SER36 (3.01 Å), and CYS45 (2.34 Å) and ARG38 (4.02 Å) and HIS374 (5.4 Å) with π-alkyl interactions in glutelin protein.

The residues involved in pro-legumin (−8.4 kcal/mol) (only D, E, F chain) from pea protein (Pro legumin) and cyanocobalamin binding were found to be ARG354, LYS357, ASN378, SER149, GLN152, ARG354, TRP355, PHE83, PRO123, ARG147, ASN150, ARG354, TRP355, LYS357, THR437, ASN438, and ASP439. The vitamin B12 and Pro legumin interaction is stabilised by 11 hydrogen bonds and 6 π-π, π-sigma, and π-alkyl interactions. The oxygen atom on P=O of cyanocobalamin formed hydrogen bonds with the NH_2_(ζ- position) group of LYS357 and NH_2_ (η-position) group of ARG354 in pea protein at distances of 3.02 and 5.17 Å. The 5,6-dimethylbenzimidazole group of cyanocobalamin is not able to form strong π-bonds with Pro legumin. On the other hand, the corrin structure of cyanocobalamin is stabilised in Pro legumin with hydrogen bonds. The residues involved in pro-legumin (−7.2 kcal/mol) (only D, E, F chain) from pea protein (Pro legumin) and aqua cobalamin binding were found to be SER149, ASN150, ARG354, TRP355, LYS357, ASN438, ASP439, ARG440, SER148, SER149, ASN150, and TRP355. The aqua cobalamin and Pro legumin interaction is stabilised by six hydrogen bonds and six π-π, π-sigma, and π-alkyl interactions. The 5,6-dimethylbenzimidazole group of aqua cobalamin forms strong π-bonds with LYS227 of Pro legumin.

A vina score is a scientific scoring function that computes the affinity or fitness of protein–ligand binding by aggregating the contributions of several individual terms [27]. Vina scores typically reflect a significant energy element in the binding of proteins and ligands. The predictions can be improved by changing the numbers of parameters that are involved in this function. A higher negative Vina score indicates strong binding stability [11,28,29,30,31,32]. All proteins showed higher binding stability with cyanocobalamin, except gluten and myoglobin.

## 5. Conclusions

In this work, the molecular recognition pattern of cobalamins based on interactions with different proteins were characterized via STD–NMR and computer simulations. Every protein has different characteristics. This study demonstrates that myoglobin shows the highest binding affinity, whereas gluten shows the lowest affinity. Casein also shows higher binding affinity for cyanocobalamin when compared with that for plant-based proteins and similarly with that for aqua cobalamin. The mechanism of cobalamin-protein affinity is a static binding mechanism with a moderate binding capability sustained by hydrogen bond, hydrophobic force, and π-π interactions. The interaction between plant-based proteins and vitamin B12 is significantly different from that of the animal proteins. Pea and rice protein shows a similar interaction pattern as with animal proteins. The results obtained by this work will clarify the potential bioavailability and biological action of cobalamins. This work suggests that chemical affinities between proteins and vitamin B12 could be used for a preliminary understanding using NMR technology before bioavailability studies can be undertaken.

## Figures and Tables

**Figure 1 foods-12-00575-f001:**
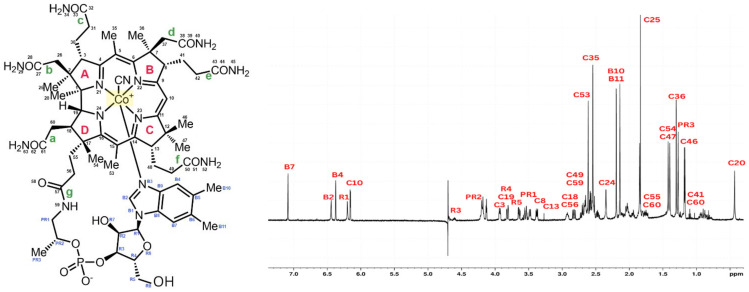
Molecular formula of cyanocobalamin (**left**). ^1^H NMR spectrum of cyanocobalamin and identification of all proton peaks (**right**).

**Figure 2 foods-12-00575-f002:**
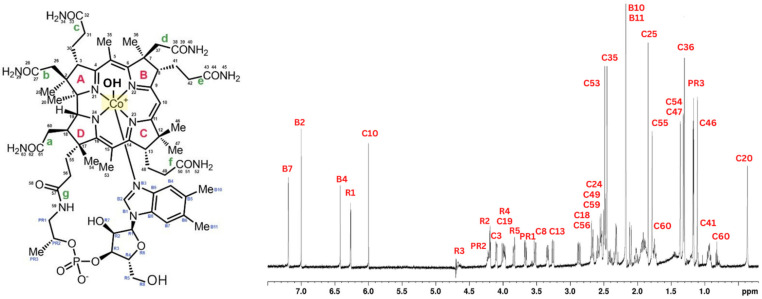
Molecular formula of aqua cobalamin (**left**). ^1^H NMR spectrum of aqua cobalamin and identification of all proton peaks (**right**).

**Table 1 foods-12-00575-t001:** Binding constants (K_α_) of different carbon atoms of vitamin B12—protein interaction interactions.

Binding Constant—K_α_ (L/mol)																		
	C20		C25		C35		C53		C10		B10/11		B4		B2		B7	
	Cyanocobalamin	Aqua Cobalamin	Cyanocobalamin	Aqua Cobalamin	Cyanocobalamin	Aqua Cobalamin	Cyanocobalamin	Aqua Cobalamin	Cyanocobalamin	Aqua Cobalamin	Cyanocobalamin	Aqua Cobalamin	Cyanocobalamin	Aqua Cobalamin	Cyanocobalamin	Aqua Cobalamin	Cyanocobalamin	Aqua Cobalamin
**Gluten**	2.88 × 10^4^	3.21 × 10^4^	3.03 × 10^4^	3.59 × 10^4^	3.7 × 10^3^	2.78 × 10^3^	1.99 × 10^4^	4.65 × 10^3^	2.77 × 10^3^	2.83 × 10^3^	3.02 × 10^4^	3.46 × 10^4^	3.16 × 10^3^	3.65 × 10^3^	3.77 × 10^3^	3.09 × 10^3^	2.76 × 10^3^	2.40 × 10^3^
**Myoglobin**	3.16 × 10^3^	4.04 × 10^4^	4.82 × 10^4^	3.04 × 10^4^	4.34 × 10^4^	4.39 × 10^4^	3.16 × 10^4^	3.51 × 10^4^	4.33 × 10^4^	3.86 × 10^4^	3.96 × 10^4^	4.67 × 10^4^	3.74 × 10^4^	4.86 × 10^4^	3.05 × 10^4^	4.37 × 10^4^	2.48 × 10^4^	4.20 × 10^4^
**Casein**	4.24 × 10^4^	3.04 × 10^4^	4.88 × 10^4^	2.71 × 10^4^	2.47 × 10^4^	4.27 × 10^4^	3.42 × 10^4^	5.52 × 10^3^	3.01 × 10^4^	3.11 × 10^4^	3.19 × 10^4^	4.72 × 10^4^	2.29 × 10^4^	3.75 × 10^4^	2.92 × 10^4^	3.11 × 10^4^	3.16 × 10^4^	3.04 × 10^4^
**Egg albumin**	2.35 × 10^4^	3.73 × 10^4^	3.04 × 10^4^	3.20 × 10^4^	2.33 × 10^4^	3.44 × 10^4^	2.43 × 10^4^	2.66 × 10^4^	1.87 × 10^4^	3.28 × 10^4^	1.61 × 10^4^	2.60 × 10^4^	2.48 × 10^4^	4.16 × 10^4^	1.60 × 10^4^	6.46 × 10^4^	2.41 × 10^4^	8.03 × 10^4^
**Pea**	7.29 × 10^3^	4.10 × 10^4^	2.96 × 10^4^	1.48 × 10^4^	1.13 × 10^4^	3.06 × 10^4^	3.85 × 10^4^	3.04 × 10^4^	1.08 × 10^4^	3.21 × 10^4^	7.65 × 10^3^	3.80 × 10^4^	1.28 × 10^4^	4.54 × 10^4^	1.09 × 10^4^	4.54 × 10^4^	1.01 × 10^4^	4.97 × 10^4^
**Rice**	3.13 × 10^4^	3.43 × 10^4^	3.84 × 10^4^	2.46 × 10^4^	3.52 × 10^4^	2.46 × 10^4^	2.21 × 10^4^	3.91 × 10^4^	2.44 × 10^4^	2.60 × 10^4^	2.49 × 10^4^	4.28 × 10^4^	2.01 × 10^4^	2.41 × 10^4^	2.31 × 10^3^	2.56 × 10^3^	4.81 × 10^4^	2.95 × 10^4^

**Table 2 foods-12-00575-t002:** K_D_ and K_α_ of vitamin B12 and protein interactions.

	K_D_ (mM)		K_α_ (L/mol)-Binding Constant	
	Cyanocobalamin	Aqua Cobalamin	Cyanocobalamin	Aqua Cobalamin
Gluten	0.23469	0.25545	4261.01	3914.70
Myoglobin	0.02911	0.02315	34,353.06	43,189.09
Casein	0.03401	0.02876	29,399.73	34,772.23
Egg albumin	0.04926	0.02527	20,301.11	39,573.24
Pea	0.10247	0.02554	9758.70	39,159.15
Rice	0.09200	0.08544	10,869.40	11,703.72

**Table 3 foods-12-00575-t003:** Vina Score and amino acids associated with vitamin B12—protein interactions.

	Cyanocobalamin		Aqua Cobalamin	
	Vina Score	Amino Acids Interacting	Vina Score	Amino Acids Interacting
Casein	−5.9	ASP184, VAL186, ILE187, ARG189, PRO214, THR219, ALA220, GLU223, ASP284, ILE351, LYS358, TYR359, PRO389, ASP390, LEU393, ASP394	−4.4	SER344, VAL345, ARG346, VAL349, ARG353, VAL373, GLN377, GLY629, ASN633, ASP634, GLU635, GLU792, GLU793, ARG794, PHE795, GLU796, GLN797, ASN798
myoglobin	−5.1	PHE44, LYS46, PHE47, ASP61, HIS65, THR68, ALA72, ALA85, HIS89, HIS98	−5.2	ARG32, LEU33, GLY36, HIS37, LYS103, GLU106, PHE107, ILE108, ASP110, ALA111, ILE113, HIS114, HIS117, GLN129
Egg Albumin	−6.5	VAL222, LYS223, TYR246, MET319, GLU320, GLU322, PHE323, LYS457, LYS464, LEU468, ARG472, ALA475, TYR480, ILE483, VAL484	−5.6	GLU43, GLU44, LYS47, MET51, TYR63, SER67, LYS68, VAL70, LYS71, ASP75, GLN78, ASN158, VAL160, SER161, HIS165
Gluten	−7.1	PRO46, ASP174, VAL177, LEU46, ARG123, VAL168, LEU170, PRO187, GLN188, PRO189, LEU190, LYS191	−7.5	HIS10, GLN44, LEU101, PHE103, PRO120, ARG123, ASP166, HIS167, VAL168, GLU169, LEU170, SER171, GLU178, VAL179, HIS180, SER181, GLY182, VAL183, CYS184, THR185, ASP186, PRO187, GLN188, PRO189, LEU203, TYR228
Rice	−8.1	ARG199, HIS200, ARG201, PHE203, PHE204, ILE211, LEU215, GLU219, ASP222, SER224, ASN226, VAL227	−6.7	TRP34, GLN35, SER36, SER37, ARG38, ARG39, GLY40, SER41, GLU44, CYS45, ARG46, PHE47, CYS78, THR79, TYR190, ILE371, ASN372, HIS374, ASN391, GLN411, HIS412, HIS413

## Data Availability

The data presented in this study are available on request from the corresponding author.

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
