# Peer review of "Molecular Recognition Patterns between Vitamin B12 and Proteins Explored through STD-NMR and In Silico Studies"

_foods, 2023, doi:10.3390/foods12030575_

Round 1

Reviewer 1 Report

I suggest minor revision and the incorporation of all important points.

The authors have chosen a very nice and interesting topic for their work. However, I suggest the authors add Infrared experiment and if possible, Raman spectroscopy in order to discuss the main changes of docking process.

Regarding Molecular docking the authors said that molecular docking simulation at 298 K was obtained by CB-Dock2 software freely available at http://cao.labshare.cn/cb-dock/. CB-Dock2 is an upgraded version of the protein-ligand blind docking. The authors should insert more detail about the time used during the simulation.

Figure 1. Molecular formula of cyanocobalamin (left). 1H NMR spectrum of cyanoco-

172 balamin and its identification of all proton peaks(right). Please put the NMR spectra in the same scale and the figures molecular formula of cyanocobalamin (left) need to separated. They are overlaid.

Please! Check the English! There are few mistakes throughout the manuscript which need to be corrected as well as some simple grammatical errors to be ironed out.

Use the article “the” before the chemical formula throughout the manuscript.

Please format this paper! There are others mistakes to be formated in the manuscript .

Reviewer 2 Report

The paper investigates the effects of different protein on their binding affinity with cyanocobalamin. The significance of the article is prominent. Some contents should be modified as this:

1. In line 188-189, the position of molecular formula and NMR spectrum should be above and down.

2. In line 229-231, why authors select those carbon to express binding constant?

3. The significance of the article's findings is not clarified.

4. Authors should clearly state the purpose and significance of different protein and vitamin B12.

Reviewer 3 Report

This work investigated the interaction between cyanocobalamin / aqua cobalamin and different proteins by STD-NMR along with molecular docking. It is interesting to carify the molecular interaction patterns between vitamin and proteins, which would help to understand the bioavailability and quantity of the vitamin in plant-based products. In addtion, NMR technology is proposed to link nutrients' chemical and physiological properties.
